# The Future of Food: Understanding Public Preferences for the Management of Agricultural Resources

**DOI:** 10.3390/ijerph18136707

**Published:** 2021-06-22

**Authors:** Erika Allen Wolters, Brent S. Steel, Sydney Anderson, Heather Moline

**Affiliations:** 1School of Public Policy, Oregon State University, 300 Bexell Hall, Corvallis, OR 97331, USA; bsteel@oregonstate.edu (B.S.S.); hmoline@gmail.com (H.M.); 2Public Policy and Administration, Boise State University, MS1935 Environmental Research Building, Boise, ID 83725, USA; sydneynichols@boisestate.edu

**Keywords:** food policy, food production, environmental values, environmental efficacy

## Abstract

The current U.S food system has managed to provide abundant food at a relatively low cost, even as the population increases. However, this unfettered growth is reaching maximum yields as demand for greater food production competes with other uses of agricultural lands. Extant ecological factors such as water scarcity are reducing food productivity, and competition for resources to produce food is becoming more apparent. This research examines public policy preferences of U.S. west coast citizens for the management of agricultural resources through the use of random household surveys. Results suggest overall support among respondents for food policies using regulatory, tax incentive, and voluntary outreach approaches. Multivariate analyses revealed that some social-demography, knowledge, environmental values, political ideology, and environmental efficacy variables were significant predictors of public opposition and support for food policies.

## 1. Introduction

Total food production is projected to increase substantially through 2050 to meet the needs of a growing and increasingly affluent population [1,2]. The current U.S. food system, through many technological advances, policies, market forces, and other drivers, has managed to provide abundant food at a relatively low cost, amid a growing population [3]. The food system includes all processes and infrastructure involved in feeding a population: growing, processing, packaging, distributing, storing, preparing, serving, and disposing of food. It is deeply interrelated with land, water, energy, society, markets, and government [4,5]. The U.S. food system is dynamic, adapting to changing demographics and consumer preferences—including ideas about health, society, economic conditions, and environmental concerns—and advances in science and technology [6]. It is complex, with markets that function at global, national, regional, and local levels. It has incorporated a diversity of policy interventions, including subsidies, regulations, taxes, mandates, and quotas [7]. In addition, it has many actors with varying needs, perceptions, and values. Evaluating the food system requires assessing the health, environmental, social, and economic effects that policies, products, and technologies can have [6].

The food system and agriculture are deeply interrelated with land, water, and energy [8]. Agriculture in the U.S. is an energy-intensive sector and a major user of ground and surface water [9]. In 2012, U.S. agriculture consumed as much energy as the state of Utah [10] through both direct and indirect uses. Direct energy use in agriculture is primarily from fossil fuels used to operate automobiles and machinery for preparing fields, planting and harvesting crops, applying chemicals, and transporting inputs and outputs to and from the market. Electricity is used largely for lighting, heating and cooling, and operation of in-farm equipment. Indirect energy is consumed off the farm for manufacturing fertilizers and pesticides [11], with the production of fertilizer being “extremely energy-intensive, requiring large amounts of natural gas” [10]. A study on food systems estimates that it is possible to reduce fossil energy by about 50% through appropriate technology changes in food production, processing, packaging, transportation, and consumption [12]. Currently, there are U.S. public programs that promote renewable energy options and recommendations in conserving fuel and energy for farms, efficient use of fertilizers and pesticides, soil conservation measures, and energy-efficient methods of growing and transporting foods [13], but those programs will need high levels of participation to curb both climate impacts as well as address air and water quality issues associated with traditional agricultural practices.

Agriculture uses an enormous amount of water, with about 80% of ground and surface water use attributed to the agricultural sector [9]. In many places in the western U.S., where drought is common, incentives for conservation compete with maintaining water rights, specifically prior appropriation where water rights holders must use their allocated amount of water or risk losing it. This disincentivizes water conservation for agricultural users and, particularly in the West, has exacerbated water scarcity and is depleting ground water resources. In addition, the large-scale application of irrigation water has increased salinity in the soil, presenting unique challenges to the continued use of land for food production. Efficient irrigation systems and better water management practices are needed to secure the long-term sustainability of irrigated agriculture while maintaining farm profitability [9]. Recommendations at the farm level include efficient irrigation systems combined with good farming practices and the use of soil and plant-moisture-sensing devices. On a regional level, recommendations include the optimization of water allocation and a continuous knowledge-exchange system to help farmers and resource managers make irrigation decisions with a greater sense of shared responsibility across the entire water supply-chain [14].

Although food production and distribution has dramatically increased, it has unanticipated repercussions. The goal of food sufficiency, or the ability to produce enough food to feed most people, has given rise to an “agri-food system that is characterized by the power and growth of transnational corporations” [15]. Specifically, the push for food sufficiency has been related to a deterioration of the environment as well as an overall reduction in nutritious food. The result being that the environmental effects of food systems are intertwined with health, social, and economic domains as well as depletion of natural resources that can disturb the ecosystem balance [16]. It also involves the use of environmental contaminants such as pesticides and nitrogen that pollute the natural environment and may be harmful to human health [5,17]. Promoting food production alternatives as well as soil, water, and energy conservation, and minimizing nutrient and chemical applications, are among management practices expected to mitigate the adverse effects of agricultural production [16].

The globalized food system has been associated with downturns in social, environmental, and human health, and a growing interest in local foods in the U.S. is the result of consumer concerns about environmental and community impacts [17,18]. As a result, alternative food systems have arisen as a social movement holding food values that oppose poor quality or low environmental standards often associated with a globalized food system [19]. Thus, food sovereignty, or the production of culturally relevant and nutritious food, has come to the forefront of food policy discussions, as evidenced by the Alternative Farming Systems Information Center (AFSIC), a USDA program that specializes in identifying resources about sustainable, alternative food and agricultural systems, and crops and livestock. Local food systems, a topic covered by the program, have a growing importance in the U.S. West. Local and regional food systems are generally defined as “agricultural production and marketing that occurs within a certain geographic proximity, involving certain social or supply chain characteristics—such as small family farms, urban gardens, or farms using sustainable agriculture practices” [20]. The reported beliefs associated with locally sourced foods over commercial-conventional stores include beliefs that local foods are fresher and higher in quality, and healthy, as well as having environmental safeguards, promoting animal welfare, and prioritizing the well-being of farmers and the fomenting of personal relationships across the supply chain [19,20].

Food and agricultural policies are aimed at influencing the behavior of the agricultural sector and agricultural markets. The intervention of government in agriculture over the years has seen a variety of policies aimed at lowering food prices, increasing food production, and increasing national and global markets [21]. In the U.S., public opinion has substantial impacts on public policy [22], and the last decade has seen a favorable and growing public support for sustainable agricultural practices and local food markets [23]. In this regard, the 2018 Farm Bill saw significant shifts for local/regional food systems, with new programs supporting the local agricultural market, urban agriculture, and studies on food waste [24]. In addition, there is growing support for direct-to-consumer access to regionally produced food, markets, and business engaged in local foods such as farmers’ markets, roadside farm stands, and intermediated outlets such as farm-to-school programs, community kitchens, local retailers, school gardens, urban farms, and community gardens [23]. According to Vasi, the local food market is a “moralized market”, where people combine economic activities with their social values [24]. Local food markets are reported to develop in areas where residents had a strong commitment to civic participation, health, and the environment [24]. Many families living in the city would like to grow some of their own fruits, vegetables, herbs, and flowers either to save money on their food bills, or for freshness, flavor, and wholesomeness of home-grown produce. To this end, community gardens offer neighborhood space to garden.

This paper seeks to explore public opinion further regarding food policy preferences in the Western U.S. states of Washington, Oregon, California, and Idaho through the analysis of random household survey research. Specifically, the paper investigates support for different food policies that includes regulatory, tax incentive, and voluntary outreach approaches. In order to understand food policy preferences more fully, we examine social-demography, knowledge, environmental values, state residency, political ideology, and environmental efficacy variables as potential predictors of public opposition and support for food policies. Understanding the public’s opinion on agriculture practices and policies for the U.S. food system can lead to efficient regulations moving towards more sustainable practices, particularly in the Western U.S. where agriculture is a predominant industry.

## 2. Background

Food production in Western states is both a dominant land use issue and a key component to state economies. Washington, Oregon, California, and Idaho each significantly contribute to food production, and will continue to serve an important role as demand for food increases. However, agriculture is an evolving industry driven by consumer preference and environmental factors. As the agricultural industry grapples with increased water scarcity in the West, and related extant factors such as energy use, water quality, and soil erosion, food policies will continue to evolve to meet these challenges while providing an ample food supply.

However, policy necessitates support from the public; thus, for an environmental policy issue to be salient to an individual, it can be assumed that the individual is both aware of the issue and is concerned or interested in mitigating the issue. As such, many studies have sought to identify social-demographic variables that are more aligned with support of pro-environmental policy instruments (many focused on climate change). Examination into social-demographic variables and environmental support have met with some inconsistent findings (for a review, see Daniels et al., 2012) [25]. Even so, there appears to be a trend of increased environmental support among younger people [26,27,28,29], females [30,31,32,33,34,35], and people with higher levels of formal education [28,36,37].

More consistent support for pro-environmental actions, beliefs, or policies center on political ideology, environmental values, and personal efficacy. Liberals generally hold stronger environmental views when compared to conservatives [35,38]. In addition, individuals who identify more strongly with environmental values (as measured by the New Ecological Paradigm (NEP)) [27,39,40] and people with a greater sense of personal efficacy (feeling their actions can have a positive impact) [41,42] are more likely to report stronger environmental beliefs.

Research suggests that public knowledge of an issue can also directly impact their support for policies. Thorvaldson, Pritchett, and Goemans (2010) [43] found that an understanding of water concepts was positively related to willingness to pay. Essentially, research posits that the greater the knowledge of an issue, the more likely people will support policies that address the issue.

Social identity theory posits that people’s perceived identity is innately connected to social groups that they belong to or identify with, which then can impact policy preferences aligned with their preferred groups. Further, between groups there can be discrimination and a tendency to perceive the other group in a negative way [44]. However, a 2016 study found that across political party lines, there was high support and perceptions of farmers [45]. Relatedly, a 2020 Gallup Poll found that farming and agriculture were the top-ranked industries in the U.S., with 69% of respondents holding positive views of farming and agriculture [46]. People identify with many social groups, either personally or more broadly, and simply view another group as positive. Research consistently shows support for farmers and agriculture suggesting that even if people do not personally engage in farming or agriculture, they identify with the positive impacts of food production.

Studies examining food preferences have found increasingly positive views toward locally produced foods with the perception that these foods are more sustainable [47]. In terms of buying local, studies have found that females [48] and people with higher incomes [49] were more likely to buy locally produced food. Relatedly, another study found that people with higher levels of formal education and younger people were more likely to purchase organic foods [50]. Further, community gardens are often supported for the ability to produce fresh, local food inexpensively [51], and for their overall positive impact on surrounding communities [48]. For these reasons, the California Healthy Cities and Communities (CHCC) program has facilitated numerous community gardens in cities throughout California [52]. A study by Foltz, Harris, and Blanck [53] explored the social-demographics of individuals supporting increased access to and funding of farmers’ markets, community gardens, small stores, and school cafeterias serving local foods. The highest support was among females, younger people, and people who were low-income, with low-income individuals more likely to support community gardens.

Demand for locally produced food has increased alongside a desire for food produced through more environmentally friendly means [54]. Some states, such as Hawaii, have responded to the consumer demand by providing an organic farming tax credit [55]. Additionally, in 2016, the USDA allocated $47 million in agricultural investments for water and energy conservation [56].

Biofuel production initially gained support due to the purported reduction of greenhouse gas emissions from biofuel as compared to fossil fuels. However, tradeoffs between food production and biofuel production are becoming more contested. Biofuels require large swaths of land, land that can be used for food production and carbon sequestration. Further, biofuel production (particularly sugar beets or corn ethanol) can create both water and air pollution, and potentially drive-up food prices. However, biofuels can provide a lower carbon energy source. While most Americans favor federal support for biofuel (ethanol) research [57], liberals are more likely to see biofuels as a positive and support more use and production of biofuels [58]. A study by Delshad et al. (2010) [59] conducted focus groups in Indiana and found that while a narrow majority favored biofuels, many more respondents preferred energy conservation or other renewable energy sources such as wind and solar.

It is known that the wide-scale application of pesticides and fertilizers has led to water pollution, health concerns, species loss, soil degradation, and habitat modification or loss [60]. According to a 2018 Pew Research Center study, a majority of Americans (79%) feel that “fruits and vegetables grown with pesticides” have “some” or “a great deal” of health risk over a person’s lifetime. Further, 51% of Americans feel that the food they are exposed to every day with added hormones, pesticides, antibiotics, etc. “pose a serious risk to their health” [61].

Geography can also influence policy preferences regarding food production. The area of this study includes four western states: California, Oregon, Washington, and Idaho. Agriculture in the western U.S. is the major user of ground and surface water, accounting for 97% of water withdrawals in Idaho [62], 80% in Washington [63], 77% in California [64], and 79% in Oregon [65]. The Columbia and Snake River Basins of the U.S. Pacific Northwest and the California Central Valley are among the top U.S. irrigated regions for crop production [9]. In terms of energy, Washington, Oregon, and California incorporate 25% of the nation’s agricultural electricity costs [11]. Feeding the U.S. population requires an estimated 15 percent of the total annual U.S. energy, mainly from fossil fuel sources, with 93% of fuel used for food production coming from fossil fuels [3].

### State Case Studies

California: California leads the nation in agricultural production and revenue. It accounts for 13% of farm commodity production nationwide [66], and in 2018 generated $49.8 billion in cash receipts [67]. California’s organic agricultural production also leads in the nation, producing over 40% of organic food in the U.S. and in 2019 totaling $10.4 billion in sales [67]. In 2014, the agricultural industry reportedly employed about 829,000 workers [68]. California is also home to the farm to fork (table) movement inspired by Alice Waters in the 1970s and originated the concept of locavore (only eating foods produced within a 100-mile radius) [69]. Further, California law has taken a step beyond the Clean Water Act (CWA) by including agriculture as a water pollutant (notably, there has yet to be a limit placed on agricultural chemical application) [70].

Oregon: Oregon’s agricultural industry is the second largest contributor to the state economy with cash receipts in 2018 valued at $4.9 billion [68]. With over 34,200 farms in Oregon, and about one in nine people employed by the agricultural industry [71], agriculture is an important economic driver in the state. Both Oregon and California also have a strong commitment to farm to school programs, with over 50% of school districts in each state participating in the program [72] by providing fresh, locally produced foods directly to schools. Additionally, about 19% of farms in Oregon sold their products directly to consumers via farmers markets and Community Supported Agriculture (CSA) programs [73].

Washington: In 2018, Washington’s agricultural cash receipts came in at $9.5 billion. With over 37,790 farms and employing about 164,000 people [74], Washington’s agricultural industry represents a robust economic sector in the state. Further, Washington is number one in the nation in the production of several crops, including apples and sweet cherries [75]. In 2016, Washington had 677 registered organic farms, placing the state in the top ten states with the most organic farms [76].

Idaho: Idaho agriculture is the largest economic contributor to the state economy [77], with 2018 cash receipts at $7.5 billion [67]. More farms in Idaho are now being certified organic; the state ranks sixth nationally in acreage for organic farms and is a top producer of organic crops that include potatoes and dairy products [78]. In 2017, the agricultural sector employed 69,800 workers, and more broadly with agribusiness providing $9.6 billion or 13 percent of Idaho’s GDP [78].

Social-demographics, environmental values, efficacy, residency, and ideology can all have an impact on support for, or opposition to, food policies. The goal of this paper is to determine which variables align more with support or opposition to food policies centered on regulatory, tax incentive, and voluntary outreach options in order to identify the potential for building policy support for more sustainable food policies.

## 3. Methods

This study used an online (Qualtrics (Qualtrics XM, Seattle, WA; Provo, UT, USA) and mail-in survey of residents from California, Oregon, Washington, and Idaho. The survey was conducted in the Spring of 2018 and employed a modified version of Dillman’s [79] tailored design method for both the online and mail-in survey. Following Dillman’s [79] method, respondents would receive up to three mailings. First, respondents were sent a postcard notifying them of the survey and providing an online option for completing the survey. Then, a physical copy of the survey was sent to residents with the option of completing the paper survey or providing an online option. A final reminder survey was sent to residents that had not completed either the online or mail-in survey. All mailings provided a description of the project, and a first class, pre-paid return envelope.

The participant sample was selected from a national sampling company using random address-based sampling (ABS) from the U.S. Postal Service’s computerized delivery sequence file (CDS). A total of 4695 valid residential addresses in California, Oregon, Washington, and Idaho were provided (California = 1170; Oregon = 1173; Washington = 1177; and Idaho = 1175) (see Table 1). Response rates varied slightly between states, with the lowest response rate from California (37.2%) and the highest from Oregon (40.5%). More California residents filled out the online survey (31.7%) compared to the other states (Idaho residents were least likely to fill out the online survey with only 18.9% opting for the Qualtrics online survey).

## 4. Findings

To assess public food policy preferences in the California, Idaho, Oregon, and Washington states, respondents were asked their level of opposition or support for seven different food policies including tax incentive policies, regulatory policies, and voluntary outreach policies. These policies are from a study by Portney et al. [80] that examined public preferences for food-water-energy policies in the U.S. The lead-in question utilized in the survey was: “A number of policy options have been proposed to manage agricultural resources. Please indicate your level of opposition or support for each of the following options”. The response categories provided are as follows: 1 = strongly oppose; 2 = oppose; 3 = neutral; 4 = support; 5 = strongly support.

Table 2 provides mean scores and F-tests for each of the seven food policies by state. For the three tax incentive policies that would encourage sustainable food production, mean scores in all four states indicate that there is more support than opposition for each of the policies; however, all three F-tests are statistically significant indicating some variation among the states. For the first policy, “give tax incentives for farmers to reduce the use of fertilizers and pesticides”, Washington respondents were the most supportive (mean = 4.11), while California respondents were supportive but less supportive (mean = 3.66). Means scores for Idaho (mean = 3.78) and Oregon (mean = 3.86) are between the mean scores for California and Washington.

The second tax incentive food policy in the survey is: “provide tax incentives for farmers to use more energy efficient methods of growing and transporting food”. Similar to the previous tax incentive policy, mean sores in each state indicate that more respondents are supportive than opposed to the policy. Moreover, similar to the previous policy, Washington respondents were significantly most supportive (mean = 4.15), and California respondents were supportive but less so than the other states (mean = 3.67). Both Idaho and Oregon respondents also had high levels of support for tax incentives for more efficient energy methods with mean scores of 4.01 and 3.96, respectively.

The third tax incentive policy concerns providing “tax incentives for farmers to use more water efficient methods of growing food”. Once again, mean scores in all four states indicate that there is more support than opposition to the policy, with Washington again having the highest mean score (mean = 4.19) and California slightly yet significantly lower than the other states (mean = 3.83). As with the first two policies, Idaho (mean = 4.09) and Oregon (mean = 3.98) fall in between the other two states.

Turning now to the two regulatory agricultural policies included in the survey, we find that respondents in all four states are more likely to support than oppose the first policy of “require that farmers use soil conservation measures”. Respondents from Oregon and Washington were the most supportive with mean scores of 3.84 and 3.85, and mean scores of 3.64 and 3.63 for California and Idaho respondents. For the second regulatory policy—“limit the amount of land that can be used to grow crops for biofuels rather than food”—mean scores trend toward support in all four states, but not as supportive as the previous regulatory policy. California respondents exhibited the lowest level of support (mean = 3.10), followed by Idaho respondents (mean = 3.22). Oregon and Washington respondents were more supportive than California and Idaho with mean scores of 3.40 and 3.47, respectively.

The final set of food policies concern voluntary outreach policies. The first policy option asked respondents their level of opposition or support for “provide space free of charge for community gardens”. As with many of the previous proposed policies, mean scores indicate that there is much more support than opposition in all four states with Oregon and Washington having the highest means scores at 4.05 and 4.02, respectively, and California and Idaho with mean scores of 3.71 and 3.93. For the second voluntary outreach policy—conduct campaigns to encourage buying locally grown foods—respondents in all four states were also more supportive than opposed with mean scores for Idaho, Oregon, and Washington above 4.0 (“agree”), and California being 3.81.

## 5. Independent and Control Variables

Social-demography, environmental values, political ideology, environmental efficacy, group identity, and a knowledge quiz are used in the forthcoming multivariate analyses as predictors of public support or opposition to the seven food policies discussed above (see Table 3). Age was an open-ended question allowing respondents to provide their exact age, with an average age of 51.5 years for the combined samples (the question used was: “What is your current age in years?”). Gender was coded as a dichotomous dummy variable (1 = female, 0 = male), with a mean of 0.504 (slightly more females participated in the survey) (the question used was: “Please indicate your gender”, with response categories of 1 = female, 2 = male, 3 = prefer not to say”). Education and annual household income were multi-categorical response formats with a mean score of 4.80 for education (“Some college, no degree”), and the mean score of 5.88 for household income ($50,000–$74,999 category). (For education the question and response categories used were: “What is your level of formal education?” 1 = less than high school (grades 1–8), 2 = some high school (no degree), 3 = high school graduate, 4 = some college, no degree, 5 = two-year associate college degree (e.g., AA), 6 = college degree (e.g., BA, BS, AB), 7 = some postgraduate schooling (no degrees), 8 = postgraduate/professional degree (e.g., MA, JD); for income the question and response categories used were: “Which category best describes your household income (before taxes) in 2017?” 1 = less than $10,000, 2 = $10,000–$14,999, 3 = $15,000–$24,999, 4 = $25,000–$34,999, 5 = $35,000–$49,999, 6 = $50,000–$74,999, 7 = $75,000–$99,999, 8 = $1,000,000–$149,999, 9 = $150,000–$199,999, 10 = $200,000 or more).

A knowledge quiz was constructed to ascertain how informed respondents are concerning food, water, energy nexus issues. Five statements were posed to assess knowledge, and respondents could choose whether the statement was “Accurate”, “Inaccurate”, or “Don’t Know”. Correct answers were coded as a 1, and wrong answers and “don’t know” were coded as a 0. Then, the answers were summed up for a total quiz score ranging 0 = no correct answers to 5 = five correct answers. The average respondent got 2.56 answers correct, which is a little over half of the questions answered with correct answers. The quiz comes from a modified version of the Portney et al. [80] survey. “Which of these statements do you believe is accurate or inaccurate?” The statements and correct answers were: (A) Using hydraulic fracturing to remove natural gas from the ground uses significant amounts of water (accurate); (B) Periods of drought can mean that an individual power plant cannot make as much electricity (accurate); (C) Recycled water cannot be safely used to grow food (inaccurate); (D) Corn used as ethanol fuel gives cars better gas mileage than gasoline; (E) Crop irrigation in the U.S. uses more groundwater than all other uses combined (accurate). Response categories included: accurate, inaccurate and don’t know. Correct responses were given a 1 and wrong responses and don’t know were given a 0 for the additive quiz score.

Environmental efficacy was ascertained utilizing a four-question Likert scale response option (1 = “Strongly Disagree” to 5 = “Strongly Agree”) that asked respondents if they believed their own behaviors can positively influence the environment and their willingness to engage in environmentally responsible behaviors. The environmental efficacy index included four questions that captured respondents sense of personal efficacy by asking level of agreement with the following four statements: (1) I feel that my own personal behavior can bring about positive environmental change; (2) I would be willing to accept cuts in my standard of living, if it helped to protect the environment; (3) I would be willing to support higher taxes, if it helped to protect the environment; and, (4) I would be willing to sacrifice some personal comforts in order to conserve resources. A composite efficacy scale was created ranging from 4 = low efficacy to 20 = high efficacy. The mean score for all four states was 14.6, indicating that the average respondent was relatively efficacious. Cronbach’s alpha for the scale is 0.804.

Political ideology was measured on a nine-point scale with 1 = “Very Liberal” to 9 = “Very Conservative” with a mean score of 4.68 for all four case study states, indicating that the average respondent was overall moderate with a slight liberal leaning (the question and response categories were: “On domestic policy issues, would you consider yourself to be?” 1 = very liberal to 9 = very conservative, with 5 = moderate). Given that California, Oregon, and Washington are liberal “blue” states, this finding is not surprising. Environmental values were assessed using six statements from the New Ecological Paradigm with respondents indicating their level of agreement using a Likert scale (1 = “Strongly Disagree” to 5 = “Strongly Agree”). The NEP is one of the most utilized indexes to assess environmental values in social science research. The following NEP statements were used in this study [27]: (1) The balance of nature is very delicate and easily upset by human activities; (2) Humans have the right to modify the natural environment to suit their needs; (3) We are approaching the limit of people the earth can support; (4) The so-called “ecological crisis” facing humankind has been greatly exaggerated; (5) Plants and animals have as much right as humans to exist; (6) Humans were meant to rule over the rest of nature. After reverse recoding of items 2, 4, and 6, the responses were summed to produce an overall index, with possible scores ranging from 6 to 30, with low scores indicating more human centered (human domination of nature) values and higher scores indicating more pro-ecological (nature for nature’s sake) values. The scale has a Cronbach’s alpha of 0.766 indicating that respondents were relatively consistent in their responses. The mean score was 20.73 indicating respondents leaned more toward a pro-environmental position.

Lastly, respondents were asked to what extent they identified with various groups with 1 = “Not At All” to 5 = “Very Strongly”. The group identity question included 7 groups that respondents could identify with. For this study, only farmers/ranchers and environmentalists are relevant groups. The question asked respondents “To what extent do you identify with each of the following groups?” Response categories included 1 = Not at all; 2 = Slightly; 3 = Moderately; 4 = Strongly; and 5 = Very strongly. Respondents were slightly more likely to identify with farmers/ranchers with a mean score of 3.26, compared to a mean score of 2.96 for environmentalists, which is almost exactly a “moderately” identification.

## 6. Multivariate Analyses

To assess the effects of the various independent and control variables presented in Table 3, we employ ordinary least squares (OLS) estimates. Table 4 presents results for the impact of the social-demographic variables, food-water-energy quiz, environmental efficacy index, the NEP and political ideology, and group identification with farmers/ranchers and environmentalists for the three tax incentive food policies. The models presented in Table 5 similarly examine the effect of these variables for the two regulatory food policies, and Table 6 presents the multivariate results for the two voluntary outreach policies. Respondents from all four study states are included for each of the OLS models presented in the three tables.

All three models in Table 4 have statistically significant F-tests at the 0.001 level, indicating that each model provided a good statistical fit. An adjusted R^2^ is also provided for each model to indicate what percent of the variation in the dependent variables each model explains. All three R^2^ coefficients are relatively strong for public opinion data with a 0.375 value for the “reduce fertilizers and pesticides” model, followed by a 0.345 value for the “use more water efficient methods” model, and a 0.293 value for the “use more energy efficient methods” model [81].

For the sociodemographic variables included in all three models, age had no statistically significant effect for any model. However, gender had a statistically significant effect for all three models with females significantly more supportive than males to support the use of tax incentives to have farmers reduce fertilizers and pesticides, to use more energy efficient methods, and to use more water efficient methods. This is consistent with much of the previously cited literature that females are more environmentally oriented than males. Education had a significant effect for two models—the tax incentives to use more energy efficient and water efficient methods. Those respondents with higher levels of formal education were significantly more supportive of these policies when compared to those with lower levels of education. Household income also produced statistically significant results for all three models. Those respondents from higher income households were significantly more supportive of tax incentives than those from lower income households.

The food-water-energy quiz, as an indicator of respondents’ issue knowledge, had no significant effect in any of the models. However, the environmental efficacy index did have a significant impact in all three models. Those respondents with higher levels of environmental efficacy were significantly more likely than those with lower levels to be supportive of tax incentives food policies to encourage a reduction in the use of fertilizers and pesticides, and to encourage the use of more energy and water efficient methods to grow food. Therefore, if one feels as though their own personal behaviors and willingness to sacrifice can contribute to environmental protection, they are also supportive of tax incentive food policies to encourage farmers into environmentally food production as well.

The next set of variables in the models concern environmental values and political ideology. Interestingly, ideology had no significant effect in any model. This is surprising given that ideology has been found to be an important predictor of the public’s preferences concerning taxes [82]. However, the NEP was statistically significant in all three models. As expected, those respondents with higher scores on the NEP were significantly more likely to support the use of tax incentives than those with lower scores, which means that higher levels of support for environmentalism were associated with higher levels of support for tax incentives to encourage farmers to use more environmentally responsible food production methods.

The final two variables in the models concern group identification—that is groups that the respondent identifies with. Group identification with farmers/ranchers was not statistically significant for any of the three models. However, identification with environmentalists did have a statistically significant effect for two models—tax incentives for reducing fertilizers and pesticides and for using more energy efficient farming methods. As would be expected, those respondents that identify with environmentalists were significantly more supportive of tax incentives when compared to those respondents who were less likely to identify with environmentalists.

Table 5 presents OLS models for the two regulatory food policies that would require soil conservation and limiting biofuels for food production. Both models had statistically significant F-tests, and the adjusted R^2^s were also reasonably large for public opinion data at a value of 0.359 for requiring farmers to use soil conservation measures, and a value of 0.224 to limit the amount of land used for biofuel production. Compared to the tax incentive policy models in Table 4, the social-demographic variables overall had less of an effect for the regulatory models with age producing the only statistically significant result. Older respondents were significantly more likely than younger respondents to support both policies. This is an interesting finding given that previous research has found that young cohorts are more likely to express environmental concern than older cohorts [83,84]. However, it may be the nature of the regulatory approach that did not resonate with the youth when compared to older respondents.

The next two variables in the regulatory models include the food-water-energy quiz and environmental efficacy. Quiz had a statistically significant effect for both models. Those respondents with higher quiz scores were significantly more likely than those with lower scores to require farmers to use soil conservation methods and to limit the amount of land that farmers can use to grow biofuels crops. Efficacy had a statistically significant effect only for the model requiring soil conservation methods. Those respondents with higher levels of environmental efficacy were significantly more likely to support soil conservation efforts when compared to those with lower levels of efficacy. So, those respondents who exhibited higher levels of knowledge, and those who were willing to participate in environmental behaviors, were more supportive of requiring farmers to use soil conservation methods than those with lower levels of knowledge and levels of environmental efficacy.

Political ideology and environmental values as measured by the NEP are the next two variables in each model. Similar to the results in Table 4, political ideology did not have a statistically significant result in either model, which is once again an interesting finding as conservatives are typically against regulatory approaches to policy, while liberals are more supportive. However, the NEP is statistically significant in both models, with those respondents with higher scores on the NEP being significantly more supportive of the two regulatory approaches to food policy than those with lower support for the NEP.

Group identity impacted policy preferences as well, with people identifying with farmers/ranchers being significantly more likely to oppose limiting biofuels land for food production when compared to those with lower levels of identification. As would be expected, those respondents who identify with environmental groups were significantly more likely than those with lower levels of identification to support requiring farmers to use soil conservation measures and to limit the amount of land that can be used to grow crops for biofuels rather than food. Biofuels have been increasingly criticized by environmental groups as not a suitable substitute for fossil fuels and leading to higher food prices and shortages [85].

The final table (Table 6) provides regression estimates for food policy preferences regarding voluntary outreach policies. As with the previous models in Table 4 and Table 5, F-tests are statistically significant at 0.000, and similar to the results in Table 5; the adjusted R^2^ coefficients are large with a value at 0.297 for the free space for community gardens model and 0.330 for the conduct local grown food campaigns. The social-demographic variables have mixed effects for both models. For the free space for community gardens model, only income was statistically significant with, ironically, upper income households more supportive when compared to lower income households. Income had no significant effect for the conduct locally grown model.

Three of the social-demographic variables had statistically significant effects for the conduct local grown foods campaign model. Age, gender, and education all had a significant positive effect for supporting locally grown foods campaigns. Older respondents, females, and the more highly educated were more supportive of local grown foods campaigns when compared to younger respondents, males, and those with lower levels of education. The gender and education results are expected given the previous literature review, but the impact of age was not necessarily expected, as younger cohorts have been in the forefront of the local food movement in the U.S. [86,87].

Quiz did not have a statistically significant effect in either model, but efficacy did in both models. Those respondents with higher levels of environmental efficacy were significantly more supportive than those with lower levels to provide free space for community gardens and to conduct locally grown foods campaigns. This makes sense given that if you feel your individual efforts can lead to environmentally positive results, then providing people an opportunity to have their own gardens and encouraging them to promote local foods would be a logical extension of efficacy.

Political ideology was not significant for either model, but the NEP again had a positive and significant effect in both models just as in all of the previous models. Those respondents with higher NEP scores were significantly more supportive of free space for community gardens and conducting locally grown food campaigns when compared to respondents with lower NEP scores. The NEP proved to be an important predictor of food policy preferences in this study, as did environmental efficacy.

For the group identification variables in the models, there was only a statistically significant result for those who identify with farmers/ranchers in the conduct locally grown foods campaign model. Interestingly, those who were more likely to identify with farmers/ranchers than other respondents were significantly less likely to support locally grown foods campaigns. It is hard to discern the reasoning behind this result without further in-depth research; however, it may be due to the emerging conflict between conventional (commercial) farming and the growing local organic farmers sector emerging in the western U.S. and in other regions [88]. Perhaps those that identify with farmers/ranchers are supportive of conventional farming and ranching, and not so ready to identify with the local foods movement.

## 7. Discussion

The aim of this study was to understand food policy preferences from residents in four U.S. Western states: California, Idaho, Oregon, and Washington. While there was variation in support for various policies, overall respondents in all four states were more supportive of all food policies, suggesting that there is an importance placed on food in these states, which is not surprising considering that all the states included in this study have strong agricultural ties to their state economies. Washington was consistently the most supportive of all types of policy preferences (tax incentives, regulatory policy, and voluntary outreach policies). California was supportive of both tax incentives and regulatory policies, but less so than the other states. The highest levels of support from all four states were the two food policies, “provide tax incentives for farmers to use more water efficient methods of growing food” and “conduct campaigns to encourage buying locally grown foods”.

Multivariate analyses for the policies pertaining to tax incentives found that females, those with higher incomes, people who expressed stronger environmental values (as measured by the NEP), and respondents with higher levels of personal efficacy were all statistically more likely to support all three tax incentive food policies. People with higher formal levels of education were more supportive of tax incentives to improve energy efficiency and water efficiency, and those who identified with environmentalists were more supportive of tax incentives to reduce fertilizer and pesticides and to use more energy efficient methods for food production and distribution.

Looking at support for regulatory policies, older respondents, those who identified with environmentalists, people who were more knowledgeable on the food-water-energy quiz, and people who expressed stronger environmental values were more supportive of both regulatory policies than other groups. People with higher levels of efficacy were more likely to support requiring soil conservation measures, but not limiting land for biofuels. Lastly, people who identified with farmers were more likely to support limiting biofuels land for food production. This is a particularly interesting finding, as people who identified with farmers wanted them to grow food and not grains for biofuels.

Lastly, there were some distinctions pertaining to level of support for the two voluntary policies. Only those who were more environmental and with higher levels of efficacy were more likely to support both voluntary outreach policies. Support for policies to conduct locally grown food campaigns was strongest among older people, females, people with higher levels of formal education, and people who identified with farmers. As noted earlier, females are more likely to buy locally produced food [44], as are older people [89,90] and people with higher levels of formal education [91]. Finally, buying locally produced food supports the local economy, thus providing a multiplier effect to the local economy [92], and for people identifying with farmers therefore have dual benefits of an economic benefit and support for agriculture in the region. Finally, people with lower income were more likely to support a policy of providing free space for community gardens. While there is an economic benefit to free community garden space, it also offers an opportunity to grow healthy foods and provides an environmental good, thus providing multiple advantages and incentives at a relatively low cost [93].

In sum, females, people with higher levels of formal education, people who identified more strongly with the NEP, and those who had a greater sense of efficacy were more likely to support tax incentives and voluntary outreach policies. Support for regulatory policies was more consistent with older people and people with more knowledge, as assessed through the quiz. What is important from these results is that, across all policy options, there was more support among all respondents than opposition. Variation in the types of policies people preferred was related to sociodemographic variables, values and beliefs, knowledge, and group identity. However, broad support for all the policies suggests that respondents were open to different policy options to maintain a food system that reduces environmental impacts, maintains farmable land, and provides access and education on local food production. Finally, as noted above, political ideology had no impact on policy preferences. This suggests that food policy is a cross-cutting issue where people along the political spectrum have some level of agreement. This is ideal for policy makers, in addition to the fact that farmers and agriculture are generally perceived in a positive way from people, because it allows for more public support, regardless of political identity.

## 8. Conclusions

Demand for food continues to increase, while extant issues such as climate change threaten availability and amount of water to farms, impact the cost of energy production, and strain the food system to keep up with demand amidst rapidly changing conditions. Further, food consumption patterns are shifting, with most Americans wanting to eat healthy, including organic food [94] and locally produced food [92]. Ironically, as more people are prioritizing locally produced food and organic food, many small family farms are struggling to stay in business. Between 2011 and 2018, over 100,000 smaller family farms were lost [95], suggesting that there is a real need to address food policy in the U.S. to maintain ample food production. Notably, all states benefit from the 2018 Farm Bill, particularly the Environmental Quality Incentives Program (EQIP), which provides both financial and technical support for farmers to conserve water and reduce soil erosion [96]. While the accessibility and feasibility of these efforts is not the scope of this research, it is important to recognize that there are existing structures to aid in conservation of farmland and provide tools to increase water efficiency and improve overall environmental quality [96]. However, this research highlights the more comprehensive policies that people support, ranging from financial incentives, regulation, and voluntary efforts.

Interestingly, this strong support for all of the food policies, especially among people who hold stronger environmental values, lends support to the importance of food sovereignty, specifically by the support for food policies that favor farmers, protect farmland and the environment, incentivize energy and water conservation, and encourage the purchasing of locally grown foods. The lack of impact of political ideology on support or opposition for food policies suggests that food policies occupy a unique space in the policy realm by garnering broad support across the political spectrum.

The study was limited in that the food policies were part of a broader survey examining food-energy-water issues. Therefore, the focus was not exclusively on food, but reasonably part of interrelated policies regarding the food-energy-water nexus. Future studies could focus more exclusively on food policies, allowing for more options and the level of support for specific policy preferences. While it is evident that there is growing awareness of the importance of sustainable food policies, research that informs policy development is critical to initiating policies with high levels of support in a timely manner.

## Figures and Tables

**Table 1 ijerph-18-06707-t001:** Survey Response Rates.

State:	Sample Size	Responses	Response Rate	% Online Return
California	1170	435	37.2%	31.7%
Idaho	1175	440	37.4%	18.9%
Oregon	1173	475	40.5%	24.2%
Washington	1177	454	38.6%	19.2%

**Table 2 ijerph-18-06707-t002:** Food Policy Preferences by State.

Question: A number of policy options have been proposed to manage agricultural resources. Please indicate your level of opposition or support for each of the following options (1 = strongly oppose; 2 = oppose; 3 = neutral; 4 = support; 5 = strongly support).						
		**CA**	**ID**	**OR**	**WA**	**F-Test**
		Mean (s.d.) n	Mean (s.d.) n	Mean (s.d.) n	Mean (s.d.) n	
Tax Incentive Policies:						
**a.**	Give tax incentives for farmers to reduce the use of fertilizers and pesticides.	3.66 (1.28) 435	3.78 (1.05) 439	3.86 (1.07) 474	4.11 (0.87) 454	14.14 *p* = 0.000
**b.**	Provide tax incentives for farmers to use more energy efficient methods of growing and transporting food.	3.67 (1.32) 435	4.01 (0.96) 439	3.96 (0.99) 474	4.15 (0.90) 454	16.64 *p* = 0.000
**c.**	Provide tax incentives for farmers to use more water efficient methods of growing food.	3.83 (1.27) 435	4.09 (0.96) 439	3.98 (1.02) 474	4.19 (0.884) 454	9.98 *p* = 0.000
Regulatory Policies:						
**d.**	Require that farmers use soil conservation measures.	3.64 (1.24) 435	3.63 (1.23) 439	3.84 (1.02) 474	3.85 (1.05) 454	5.27 *p* = 0.001
**e.**	Limit the amount of land that can be used to grow crops for biofuels rather than food.	3.10 (1.19) 435	3.22 (1.21) 439	3.40 (1.14) 474	3.47 (1.01) 453	9.83 *p* = 0.000
Voluntary Outreach Policies:						
**f.**	Provide space free of charge for community gardens.	3.71 (1.324) 435	3.93 (1.00) 439	4.05 (0.96) 474	4.02 (0.875) 454	9.84 *p* = 0.000
**g.**	Conduct campaigns to encourage buying locally grown foods.	3.81 (1.12) 435	4.02 (0.92) 439	4.18 (0.74) 474	4.22 (0.75) 454	19.67 *p* = 0.000

**Table 3 ijerph-18-06707-t003:** Independent and Control Variables.

Variable Name	Variable Description	Mean (s.d.)
Age	Age in years (range = 18 to 98)	Mean = 51.6 s.d. = 16.83 n = 1796
Gender	Gender dummy variable (1 = female, 0 = male)	Mean = 0.504 n = 1787
Education	Formal educational attainment (1 = less than high school to 8 = postgraduate degree)	Mean = 4.80 s.d. = 1.46 n = 1798
Income	Household income before taxes in 2019 (1 = less than $10,000 to 10 = $200,000 or more)	Mean = 5.88 s.d. = 1.80 n = 1772
Quiz	Food, Water Energy Quiz (0 = no correct answers to 5 = five correct answers)	Mean = 2.56 s.d. = 1.38 n = 1804
Efficacy	Environmental efficacy index (4 = low efficacy to 20 = high efficacy)	Mean = 14.16 s.d. = 3.94 n = 1793
NEP	New Ecological Paradigm (6 = low level of support to 30 high level of support)	Mean = 20.73 s.d. = 5.43 n = 1782
Ideology	Subjective Political Ideology (1 = very liberal to 9 = very conservative)	Mean = 4.68 s.d. = 1.25 n = 1782
Group Identity: Farmers and Ranchers	Identify with Farmers and Ranchers (1 = not at all to 5 = very strongly)	Mean = 3.21 s.d. = 1.25 n = 1792
Group Identity: Environmentalists	Identify with Environmentalists (1 = not at all to 5 = very strongly)	Mean = 2.96 s.d. = 1.31 n = 1792

**Table 4 ijerph-18-06707-t004:** Regression Estimates for Food Policy Preferences: Tax Incentives.

	Reduce Fertilizers and Pesticides	Use More Energy Efficient Methods	Use More Water Efficient Methods
	Coefficient (S.E.)	Coefficient (S.E.)	Coefficient (S.E.)
Age	0.001 (0.001)	0.000 (0.001)	–0.002 (0.001)
Gender	0.287 *** (0.043)	0.121 ** (0.046)	0.206 *** (0.043)
Education	–0.007 (0.016)	0.055 *** (0.016)	0.043 ** (0.016)
Income	0.057 *** (0.013)	0.059 *** (0.013)	0.056 *** (0.013)
Quiz	0.024 (0.016)	–0.005 (0.017)	–0.002 (0.016)
Efficacy	0.041 *** (0.007)	0.044 *** (0.008)	0.031 *** (0.007)
NEP	0.076 *** (0.005)	0.076 *** (0.006)	0.090 *** (0.005)
Ideology	–0.008 (0.012)	−0.018 (0.013)	0.003 (0.012)
Group Identify: Farmers	–0.005 (0.018)	0.033 (0.019)	0.013 (0.018)
Group Identify: Environmentalists	0.069 *** (0.020)	0.048 * (0.021)	–0.027 (0.020)
N =	1700	1700	1700
F-Test =	103.01 ***	71.510 ***	90.527 ***
Adj. R^2^ =	0.375	0.293	0.345

* *p* ≤ 0.05; ** *p* ≤ 0.01; *** *p* ≤ 0.001.

**Table 5 ijerph-18-06707-t005:** Regression Estimates for Food Policy Preferences: Regulatory Policies.

	Require Soil Conservation	Limit Biofuels Land for Food
	Coefficient (S.E.)	Coefficient (S.E.)
Age	0.007 *** (0.001)	0.008 *** (0.002)
Gender	–0.062 (0.047)	0.095 (0.051)
Education	–0.003 (0.017)	0.032 (0.018)
Income	0.010 (0.014)	0.003 (0.015)
Quiz	0.168 *** (0.017)	0.101 *** (0.019)
Efficacy	0.040 *** (0.008)	0.006 (0.009)
NEP	0.082 *** (0.006)	0.047 *** (0.007)
Ideology	0.000 (0.013)	0.016 (0.014)
Group Identify: Farmers	–0.013 (0.020)	–0.116 *** (0.022)
Group Identify: Environmentalists	0.083 *** (0.022)	0.194 *** (0.024)
N =	1700	1700
F-Test =	96.188 ***	50.052 ***
Adj. R^2^ =	0.359	0.224

*** *p* ≤ 0.001.

**Table 6 ijerph-18-06707-t006:** Regression Estimates for Food Policy Preferences: Voluntary Outreach Policies.

	Free Space for Community Gardens	Conduct Locally Grown Foods Campaign
	Coefficient (S.E.)	Coefficient (S.E.)
Age	0.001 (0.001)	0.005 *** (0.001)
Gender	0.058 (0.045)	0.131 *** (0.037)
Education	0.027 (0.016)	0.039 ** (0.013)
Income	–0.038 ** (0.013)	–0.013 (0.011)
Quiz	0.014 (0.016)	–0.018 (0.014)
Efficacy	0.080 *** (0.008)	0.072 *** (0.006)
NEP	0.054 *** (0.006)	0.049 *** (0.005)
Ideology	–0.001 (0.013)	0.007 (0.010)
Group Identify: Farmers	–0.017 (0.019)	–0.041 ** (0.016)
Group Identify: Environmentalists	0.027 (0.021)	0.001 (0.017)
N =	1700	1700
F-Test =	72.832 ***	84.594 ***
Adj. R^2^ =	0.297	0.330

** *p* ≤ 0.01; *** *p* ≤ 0.001.

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
