# Peer review of "The Future of Food: Understanding Public Preferences for the Management of Agricultural Resources"

_ijerph, 2021, doi:10.3390/ijerph18136707_

Round 1

Reviewer 1 Report

The paper “The Future of Food: Understanding Public Preferences for the Management of Agricultural Resources” deals with very interesting and important topic.

Having said that, your study is very promising, but it needs some improvements. With the intention that you can make those improvements in your future research, I will comment on some of the most relevant weaknesses found in the different sections. I hope you find those comments useful.

The introduction should include specific research objectives by specifying variables to study whose justification should be clearly stated. Please, set out the specific research objectives by finding ground for it.

The review of the literature should have culminated with a set of hypotheses perfectly aligned to the subsequent empirical evidence that has been obtained in the analysis of the results section.

Please correct the table according to the publisher's requirements (see Instructions for Authors).

In conclusion, the limitations of the study and future research developments should be indicated.

The References section is not prepared according to the Instructions for Authors. References should be described as follows (Websites):

Title of Site. Available online: URL (accessed on Day Month Year).

I hope these comments can be of help in improving the paper and encourage the authors to move forward.

Author Response

Response to Reviewers

June 7, 2021

First, thank you to the reviewers for taking the time to provide useful feedback on our paper. Your time and effort is greatly appreciated. I will now respond to each reviewer’s feedback to note the changes that were made as a result of their review.

Reviewer #1

The paper “The Future of Food: Understanding Public Preferences for the Management of Agricultural Resources” deals with very interesting and important topic.

Having said that, your study is very promising, but it needs some improvements. With the intention that you can make those improvements in your future research, I will comment on some of the most relevant weaknesses found in the different sections. I hope you find those comments useful.

1) The introduction should include specific research objectives by specifying variables to study whose justification should be clearly stated. Please, set out the specific research objectives by finding ground for it.

Response: We have now included additional information about the intent and objectives of our paper in the final paragraph of the introduction (lines 126-136). Specifically, we outline the variables that we examine in relation to support for food policies ranging from regulatory, tax incentives, and voluntary outreach. We feel this is now a more robust introduction to our research objectives.

2) The review of the literature should have culminated with a set of hypotheses perfectly aligned to the subsequent empirical evidence that has been obtained in the analysis of the results section.

Response: Thank you for this suggestion. It is not typical in our field(s) to have hypotheses in our papers, more typical are research questions. At the end of the background section, we have included a summary of the variables, goals, and intent of our paper, specifically “to determine which variables align more with support of opposition to food policies centered on regulatory, tax incentive, and voluntary outreach options in order to identify potential for building policy support for more sustainable food policies” (lines 258-262). We hope this is sufficient. Thank you.

3) Please correct the table according to the publisher's requirements (see Instructions for Authors).

Response: We have consulted the Instruction for Authors and provided revised tables according to the publisher’s requirements. Thank you.

4) In conclusion, the limitations of the study and future research developments should be indicated.

Response: The conclusion section now has a discussion on limitations of the study and future research opportunities (lines 605-611). Thank you.

5) The References section is not prepared according to the Instructions for Authors.

Response: Thank you for your discerning eye on the references, we have now aligned the references with the Instructions for Authors. Thank you again.

Reviewer 2 Report

This is a research of great interest that addresses a very up-to-date and relevant issue, not only in the United States, but in the generality of the world. It is about the role of agriculture in food production, for which there are growing demands. In addition, the research is very useful in relation to determining the role of citizens in these questions, as well as with respect to economic-financial support for large-scale and family agricultural production, etc. All this happens in a context, increasingly generalized on a planetary scale, of rising demands and requirements for this production to be efficient and ecologically sustainable. The article is well written in contrast to some texts that one has to read and reread because one does not quite understand what they mean. The introduction and background are timely and clearly synthesized and explained. In the manuscript as a whole, the paragraphs are well sequenced, in such a way that each of them leads to the next in a clear and coherent line of argument. Furthermore, the manuscript is well founded bibliographically, thus well-grounded methodologically, empirically and statistically. The discussion and conclusions at the end of the text are clearly stated and solidly supported. As such, they can be of great help to responsible policy makers charged with managing agricultural policy in the United States. 
My only suggestion is that the authors introduce into their argumentation the concepts of "Food sufficiency" and "Food sovereignty" in terms of explaining what is considered an adequate system of food production. The introduction of these two concepts, as well as a brief explanation of the differences between them, will allow the authors a better explanation of one of the key objectives of this paper; which, according to its authors, "seeks to further explore public opinion regarding food policy preferences in the Western U.S. states of Washington, Oregon, California, and Idaho." 
Once the authors have introduced the concepts of "Food sufficiency" and "Food sovereignty", I think they will be in a better position to explain paragraphs like the following in more detail: 
"More consistent support for pro-environmental actions, beliefs, or policies center on political ideology, environmental values, and personal efficacy. Liberals generally hold stronger environmental views, when compared to conservatives. In addition, individuals who identify more strongly with environmental values (as measured by the New Ecological Paradigm [NEP]) and people with a greater sense of personal efficacy (feeling their actions can have a positive impact) are more likely to report stronger environmental beliefs. " 
In this regard, I would dare to assure that those people who are most favorable to pro-environmental actions are very likely also those who see adequate food production as something closer to what is understood by Food sovereignty, while socio-political positions that are more conservative are usually those that are closest to considering as an appropriate food production system the one that ensures Food sufficiency. 
Following this same reasoning, the perspectives of small farmers and those who seek local and proximity marketing of agricultural products (a perspective to which the authors allude in note [19]), would be closer to what is understood by the search for Food sovereignty. I copy the mentioned note below:
 “Agricultural production and marketing that occurs within a certain geographic proximity, involving certain social or supply chain characteristics - such as small family farms, urban gardens, or farms using sustainable agriculture practices” [19]. 
In addition to the above, I suggest that authors consult and include in their text references from Alessandro Bonanno, a prestigious and internationally known specialist on the subject of globalization of agriculture and food. For example, they can find many of these references in Bonanno's academic profile on Google Scholar:
 https://scholar.google.com/citations?user=u57yuiQAAAAJ&hl=en

Author Response

Response to Reviewers

June 7, 2021

First, thank you to the reviewers for taking the time to provide useful feedback on our paper. Your time and effort is greatly appreciated. I will now respond to each reviewer’s feedback to note the changes that were made as a result of their review.

REVIEWER #2

This is a research of great interest that addresses a very up-to-date and relevant issue, not only in the United States, but in the generality of the world. It is about the role of agriculture in food production, for which there are growing demands. In addition, the research is very useful in relation to determining the role of citizens in these questions, as well as with respect to economic-financial support for large-scale and family agricultural production, etc. All this happens in a context, increasingly generalized on a planetary scale, of rising demands and requirements for this production to be efficient and ecologically sustainable. The article is well written in contrast to some texts that one has to read and reread because one does not quite understand what they mean. The introduction and background are timely and clearly synthesized and explained. In the manuscript as a whole, the paragraphs are well sequenced, in such a way that each of them leads to the next in a clear and coherent line of argument. Furthermore, the manuscript is well founded bibliographically, thus well-grounded methodologically, empirically and statistically. The discussion and conclusions at the end of the text are clearly stated and solidly supported. As such, they can be of great help to responsible policy makers charged with managing agricultural policy in the United States. 

1) My only suggestion is that the authors introduce into their argumentation the concepts of "Food sufficiency" and "Food sovereignty" in terms of explaining what is considered an adequate system of food production. The introduction of these two concepts, as well as a brief explanation of the differences between them, will allow the authors a better explanation of one of the key objectives of this paper; which, according to its authors, "seeks to further explore public opinion regarding food policy preferences in the Western U.S. states of Washington, Oregon, California, and Idaho." Once the authors have introduced the concepts of "Food sufficiency" and "Food sovereignty", I think they will be in a better position to explain paragraphs like the following in more detail: 
"More consistent support for pro-environmental actions, beliefs, or policies center on political ideology, environmental values, and personal efficacy. Liberals generally hold stronger environmental views, when compared to conservatives. In addition, individuals who identify more strongly with environmental values (as measured by the New Ecological Paradigm [NEP]) and people with a greater sense of personal efficacy (feeling their actions can have a positive impact) are more likely to report stronger environmental beliefs. " 
In this regard, I would dare to assure that those people who are most favorable to pro-environmental actions are very likely also those who see adequate food production as something closer to what is understood by Food sovereignty, while socio-political positions that are more conservative are usually those that are closest to considering as an appropriate food production system the one that ensures Food sufficiency. 
Following this same reasoning, the perspectives of small farmers and those who seek local and proximity marketing of agricultural products (a perspective to which the authors allude in note [19]), would be closer to what is understood by the search for Food sovereignty. I copy the mentioned note below:
 “Agricultural production and marketing that occurs within a certain geographic proximity, involving certain social or supply chain characteristics - such as small family farms, urban gardens, or farms using sustainable agriculture practices” [19]. 
In addition to the above, I suggest that authors consult and include in their text references from Alessandro Bonanno, a prestigious and internationally known specialist on the subject of globalization of agriculture and food. For example, they can find many of these references in Bonanno's academic profile on Google Scholar:
 https://scholar.google.com/citations?user=u57yuiQAAAAJ&hl=en

Response: Thank you so very much for your suggestion on food security and food sovereignty and for directing us to the work of Dr. Bonanno. We have incorporated some discussion of food security and food sovereignty in the introduction (lines 75-106) and included the work of Dr. Bonanno (line 78). We also discuss this in the context of our findings in the conclusion of the paper (lines 598-604). More importantly, this suggestion has inspired future research that more fully illuminates people’s policy preferences based on their alignment with food security and food sovereignty, opening up a number of new questions to ask in future surveys. Thank you very much for this recommendation!

Round 2

Reviewer 1 Report

After reading the revised version of the manuscript, I consider that you addressed all my remarks.